# Temperature and Humidity Sensitivity of Polymer Optical Fibre Sensors Tuned by Pre-Strain

**DOI:** 10.3390/s22197233

**Published:** 2022-09-23

**Authors:** Andreas Pospori, Andreas Ioannou, Kyriacos Kalli

**Affiliations:** Photonics and Optical Sensors Research Laboratory, Cyprus University of Technology, Saripolou 33, 3036 Limassol, Cyprus

**Keywords:** polymer optical fibre, fibre Bragg gratings, sensors, temperature, humidity, sensitivity, CYTOP, XYLEX

## Abstract

Polymer optical fibre Bragg grating (POFBG) sensors are of high interest due to their enhanced fracture toughness, flexibility in bending, and sensitivity in stress and pressure monitoring applications compared to silica-based sensors. The POFBG sensors can also detect humidity due to the hydrophilic nature of some polymers. However, multi-parameter sensing can cause cross-sensitivity issues in certain applications if the temperature and humidity measurements are not adequately compensated. In this work, we demonstrate the possibility of selectively tuning sensors’ temperature and humidity sensitivities to the desired level by applying a certain amount of fibre pre-strain. The temperature sensitivity of POFBG sensors fabricated in perfluoropolymers (CYTOP) can be selectively tuned from positive to negative values, having the option for insensitivity in specific temperature ranges depending on the amount of the applied pre-strain. The humidity sensitivity of sensors can also be changed from positive values to insensitivity. The importance of thermal annealing treatment of POFBG sensors for improved repeatability in temperature measurements is also reported. An array of 4 multiplexed POFBGs was fabricated, and each sensor was pre-strained accordingly to demonstrate the possibility of having targeted temperature and humidity sensitivities along the same fibre.

## 1. Introduction

Fibre sensors are immune to electromagnetic interference, are smaller in size and more sensitive to certain measurements when compared with conventional electrical sensors [1]. The optical fibre, used as a sensing element, can detect strain, stress, deformation, humidity, temperature, or any other quantity that can change the fibre’s physical, mechanical, or optical condition [1,2]. One benefit of fibre sensing is the multiplexing capability, which means multiple point sensors can be incorporated along the fibre length [3] or in the case of distributed sensing, the entire fibre becomes the sensing element [4]. The most common type of point sensor is the fibre Bragg grating (FBG), which is a periodic refractive index modulation structure fabricated in the fibre core [2,5]. The FBG structure reflects light at a specific wavelength, called the Bragg wavelength (*λ_B_*), and it is defined by:*m λ*_*B*_ = 2 *n*_*eff*_
*Λ*,(1)
where *m* is the Bragg order (positive integer), *n_eff_* is the effective refractive index of the fibre core, and *Λ* is the spatial modulation period. The FBG can be used not only as a wavelength filter [6] but also as a sensor by monitoring the Bragg wavelength shift that occurs during induced changes, influencing the effective refractive index (*n_eff_)* and the periodicity (*Λ*) of the grating [7]. The sensitivity of the FBG sensor is usually determined by the ratio of Bragg wavelength shift to the induced change (e.g., fibre strain) [5]. The properties of the material used to fabricate the optical fibre can affect the sensitivity and overall performance of the sensing system [8]. For this reason, researchers have shown interest in polymer optical fibre (POF) Bragg grating (POFBG) sensors, which perform better in certain applications than the already commercially exploited silica FBG sensors [8]. The POFs have enhanced fracture toughness and bending flexibility [9], making them ideal for installing or embedding the POFBG sensors in other structures for structural health monitoring [10,11]. Moreover, POFBG sensors have enhanced stress and pressure sensitivity due to the lower Young’s modulus of the polymer when compared to silica [8]. POFBG sensors fabricated in water-absorbing polymers can also detect the humidity or concentrations of water-soluble analytes (e.g., sugar solutions) [12,13].

The main disadvantage of POFs is the high optical attenuation, especially in the NIR wavelength region [14]. However, novel polymeric materials were developed recently to reduce the optical attenuation, enabling the potential of POFBG sensors to be incorporated in large-scale structures (e.g., civil infrastructures) that require several metres or kilometres of fibre [15,16]. A famous example of a high-transparent polymer is commercially known as CYTOP, which is an amorphous perfluoropolymer [17]. CYTOP-based POFBG sensors were used in the experiments presented in this work. Another drawback of POFBG sensors is their operation at much lower temperature ranges than silica-based sensors due to the distinct thermal properties of polymers [18]. Changes in polymers’ physical, mechanical, and optical properties can occur above the β-transition temperature (for some polymers, this occurs even below room temperature), in which whole side chains and some backbone atoms can be moved and rearranged in the molecular structure [19,20,21]. Exposing the POF above its β-transition and below its glass-transition temperature (also known as thermal annealing treatment) has been utilised in the past not only for extending the linear response of sensors in higher temperatures [22] but also for multiplexing purposes [23,24] and sensing sensitivity enhancement [25,26].

Since the FBG sensors can detect many parameters simultaneously, such as strain and temperature, the cross-sensitivity issues must be addressed and compensated. Depending on the application, compensation techniques are often required to extract the actual sensing parameter (e.g., temperature compensation in strain monitoring). The compensation can be challenging when the fibre’s material properties change under various environmental conditions. This is particularly true for the POF-based sensors, whose properties can be affected not only by the polymer’s chemical composition and polymerisation process but also by the fibre-drawing process conditions and the thermal history of the POF [27,28]. The polymer properties can also be changed when the fibre is strained [29,30] or when water absorption occurs [31]. The complexity of the polymer’s nature is often neglected, leading to inconsistent results and non-linear responses of POFBG sensors, especially in temperature and humidity measurements [32]. Exposing the POF above its β-transition temperature, accidentally inducing fibre strain during installation, or neglecting the water absorption in the POF can dramatically affect the response of the POFBG sensor. The Bragg wavelength shift (Δ*λ_Β_*) due to changes in humidity (*H*), temperature (*Τ*), and strain (*ε*) is defined by:Δ*λ*_*Β*_ = *λ*_*Β*_ [Δ*H* (*η* + *β*) + Δ*Τ* (*ξ* + *α*) + Δ*ε* (1 − *ρ*)],(2)
where *λ_Β_* is the initial spectral position of Bragg wavelength, *η* is the normalised refractive index change due to humidity change, *β* is the swelling coefficient related to water absorption, *ξ* is the thermo-optic coefficient, *α* is the thermal expansion coefficient due to temperature change, and *ρ* is the effective photoelastic coefficient of the material. In the case of silica optical fibres, both thermal expansion and thermo-optic coefficients are positive, and the Bragg wavelength shifts to longer wavelengths as the temperature increases. However, the thermo-optic coefficient in polymers is negative and counteracts the thermal expansion coefficient, affecting the Bragg wavelength response. The positive thermal expansion coefficient of poly(methyl methacrylate) (PMMA), the most common material used to fabricate POFs, is much smaller than the negative thermo-optic coefficient; as a result, the Bragg wavelength always shifts to shorter wavelengths as the temperature increases—indicating a negative sensitivity. It is reported that the temperature sensitivity of PMMA POFBG sensors can be significantly enhanced by applying a sufficient amount of fibre pre-strain to eliminate the counteracting thermal expansion coefficient [33]. PMMA POFBG sensors at 40% relative humidity (RH) exhibit sensitivities of −37 pm/°C when unstrained and −123 pm/°C when pre-strained [33]. At 85% RH, the temperature sensitivity increases to −52 pm/°C for the unstrained sensors and −134 pm/°C for the pre-strained sensors [33]. The humidity sensitivity of PMMA POFBG sensors is reported to be 39 pm/%RH at 25 °C, 33 pm/%RH at 35 °C, and 16 pm/%RH at 50 °C [29]. However, pre-straining the PMMA POFBG sensors reduced the humidity sensitivity to 30 pm/%RH at 25 °C, 23 pm/%RH at 35 °C, and 6 pm/%RH at 50 °C [29]. The temperature and humidity sensitivity of CYTOP POFBG sensors is reported to be 27.5 pm/°C and 10.3 pm/%RH, respectively [34]. The pre-strain effects on temperature and humidity sensitivity of CYTOP POFBG sensors were not investigated until now.

This work investigated the temperature and humidity response of CYTOP-based POFBG sensors under different fibre pre-strain values. Considering that the CYTOP’s thermal expansion coefficient (7.4 × 10^−5^/°C) is larger than its thermo-optic coefficient (−5.0 × 10^−5^/°C) [35], the CYTOP POFBG sensors can have a positive sensitivity when the fibre thermally expands freely, and a negative sensitivity when the fibre pre-strain restricts the thermal expansion [36]. Applying a fibre pre-strain will also decrease the humidity sensitivity due to the elimination of the swelling coefficient. For demonstration purposes, an array of four multiplexed POFBGs was fabricated, and each POFBG was pre-strained at a different value, demonstrating the possibility of having targeted temperature and humidity sensitivities along the same fibre.

## 2. Fabrication of Bragg Grating Sensors

The fabrication of POFBG sensors was performed as follows. A femtosecond laser system (High Q laser femtoREGEN) with 517 nm operating wavelength, 2 kHz repetition rate, 220 fs pulse duration, and 20 nJ pulse energy was utilised to directly inscribe the Bragg gratings in a custom-made optical fibre (GigaPOF Custom Fibre 250/20, Chromis Fiberoptics, Warren, NJ, USA). This custom-made POF has a CYTOP (amorphous fluoropolymer) graded-index core (diameter of 20 ± 4 μm) and a XYLEX X7200 (a blend of polycarbonate and polyester) cladding (diameter of 250 ± 5 μm). The POF was attached to an air-bearing high-precision translation stage (Aerotech), and the laser was focused through the cladding into the fibre core using a long working distance objective lens with 50x magnification and 0.42 NA (Mitutoyo, Aurora, IL, USA). Utilising the translation stage, 850 Bragg planes (width of 5 μm each) were inscribed at the centre of the core and transversely across the fibre length to create each POFBG, as illustrated in Figure 1. The inscription of each Bragg plane individually is known as the plane-by-plane Bragg grating inscription method [37]. This inscription method can be offered only by femtosecond laser systems [38]. When the Bragg planes’ size is smaller than the fibre core’s size, the number of forward-modes coupling with the backward-mode of the Bragg grating is reduced. Single-peak reflection POFBGs were already demonstrated in graded-index multimode POFs using the plane-by-plane method [15,37,39].

Two POFs were used in the experiments performed in this work. The first POF incorporated one Bragg grating structure with 4th order resonance at 1549.66 nm (Figure 2), which was initially used to characterise the strain sensitivity and the temperature response under different pre-strain values. Then, a second POF was used into which was fabricated an array of 4 multiplexed Bragg gratings (physical distance of 10 cm between each grating) with 4th order resonances at 1526.62 nm, 1550.14 nm, 1564.79 nm, and 1576.53 nm (Figure 3). The array of POFBG sensors was developed to demonstrate the possibility of having sensors in the same fibre with different temperature and humidity sensitivities selectively tuned to the desired level. After the fabrication of POFBG sensors, each POF was butt-coupled and UV glued with a silica pigtail for interrogation purposes. A Bragg grating was also inscribed through the acrylate coating of a standard silica optical fibre (SMF-28) to be compared with the response of the POFBG sensors.

## 3. Strain Characterisation

This section reports the strain sensitivity of the single POFBG sensor. The strain characterisation was performed by straining the single POFBG sensor (spectrum shown in Figure 2) in steps using a high-precision linear translation stage, as illustrated in Figure 4. Using a commercial interrogator (Micron Optics HYPERION si155), the spectral position of Bragg grating was taken within seconds after performing the step. As depicted in Figure 5, the strain response was linear without any observable hysteresis, and the calculated strain sensitivity (using linear fitting) is 1.27 ± 0.01 pm/με.

## 4. Pre-Straining Procedure

This section describes the procedure to pre-strain the POFBG sensors for temperature and humidity characterisation. While the POF was strained, a 75 × 25 × 1-mm glass slide (SuperFrost Ultra Plus, Thermo Scientific, Braunschweig, Germany) was placed under the grating structure and adhered using a UV curable adhesive (Norland Optical Adhesive 61—MIL-A-3920, Norland Products Inc., Jamesburg, NJ, USA); this adhesive was chosen as it has low shrinkage and high resiliency. The adhesive, approximately 5 ± 2 mm in diameter, was placed at the edge of the glass slide, as shown in Figure 6. The thermal expansion of the adhesive and glass slide is 2.3 × 10^−4^/°C and 8.4 × 10^−6^/°C, respectively, which can contribute to the overall thermal expansion of the POFBG sensor. The thermal expansion coefficient of XYLEX X7200 (material of fibre cladding) and CYTOP (material of fibre core) is 8.0 × 10^−5^/°C and 7.4 × 10^−5^/°C, respectively. It is noted that the sensing system presented in this work was evaluated as a whole, with the materials above having different thermal properties. Therefore, a calibration of the new sensing system will be required when using materials with other thermal expansion coefficients (e.g., different POF or adhesive), and different pre-strain values may be required to achieve the desired response of POFBG sensors.

The single POFBG sensor (Figure 2) was pre-strained to three different levels (0%, 0.04% and 0.5%) for temperature characterisation. In the case of POFBG array sensors (Figure 3), each sensor was pre-strained once but at a different value to demonstrate the possibility of having different temperature and humidity sensitivities along the same POF. The pre-strain values used for the POFBG array sensors are listed in Table 1.

## 5. Temperature Characterisation

This section describes the experiment performed to assess the temperature sensitivity of sensors under different pre-strain values. The POFBG sensors were placed in a climate chamber (Memmert HCP 108) with controlled temperature and relative humidity to monitor their response to temperature. The relative humidity (RH) was kept constant at 40 ± 1% during the temperature characterisation. The temperature was changed in steps and kept stable for more than 2 h in each step to ensure stabilised climate conditions and a minimal readout of errors from the sensors. Using a commercial interrogation system (Micron Optics HYPERION si155), the sensors were monitored by tracking the reflected Bragg wavelength peak every minute using the commercial Micron Optics ENLIGHT software. The whole reflection spectrum was also captured every 15 min in case the peak tracking failed. The temperature and relative humidity measurements were captured by the internal sensors of the climatic chamber using the Celsius software (MEMMERT GmbH & Co.KG, Schwabach, Germany).

Initially, the single POFBG sensor was characterised without pre-strain and with pre-strain values of 0.5% and 0.04%. As shown in Figure 7, without any applied pre-strain, the Bragg wavelength shifts to longer wavelengths as the temperature increases, indicating positive sensitivity of 18.2 ± 0.6 pm/°C. By applying a pre-strain of 0.5%, the Bragg wavelength shifts to shorter wavelengths as the temperature increases, showing a negative sensitivity of −69.5 ± 4.3 pm/°C. Since the sensitivity of the same POFBG sensor can be positive and negative, there is also a possibility to have insensitivity to specific temperature ranges by having the right amount of fibre pre-strain that balances the overall linear thermal expansion and thermo-optic coefficients. Figure 7 demonstrates temperature insensitivity (−5.0 ± 4.3 pm/°C) for the temperature range between 30 °C and 60 °C when the fibre pre-strain is 0.04%.

To demonstrate the possibility of having different sensitivities along the same fibre piece, an array of four POFBG sensors was fabricated (spectrum shown in Figure 3), and each POFBG was pre-strained at a different amount, as listed in Table 1. During the first temperature characterisation, it was observed that the Bragg wavelength position of each sensor did not remain at the same spectral position under stabilised environmental conditions, as depicted in Figure 8. The explanation we provide is that thermal molecular relaxation was occurring even at 30 °C, leading to the Bragg wavelength shift to shorter wavelengths due to fibre shrinkage. To test our assumption, the POF needed to be annealed and the temperature characterisation to be repeated. In the case of the single POFBG sensor, the thermal relaxation was not observed because it was previously exposed to high temperatures (up to 60 °C). During the preliminary experiments, the single POFBG sensor was placed multiple times in a climatic chamber for temperature characterisation because initially, the POF was wrongly fixed with tape on paperboard, and some amount of tension was introduced on the fibre, which led to inconsistent responses every time the experiment was repeated. Therefore, the thermal history of the single POFBG was not the same as the multiplexed POFBGs. For this reason, it was decided to anneal the multiplexed POFBGs in the climatic chamber at 70 °C temperature and 40% humidity. The profile of annealing conditions is shown in Figure 9. After thermal annealing treatment, the temperature characterisation of POFBG array sensors was repeated. Comparing the results before (Figure 8) and after the annealing (Figure 10), the response of sensors becomes more linear and stable under stabilised climate conditions. Therefore, pre-annealing this type of POF before any use in temperature monitoring applications may be a prerequisite for avoiding any fibre shrinkage during temperature exposure. In Figure 10, the data density between 34 h and 41 h is lower because the interrogator failed to detect the peak position (taken every minute) in that timeframe. The missing points have been reconstructed using the data from the reflection spectra taken every 15 min.

Figure 11 shows the temperature response of the array of the four multiplexed POFBG sensors. A silica-based Bragg grating sensor was also placed in the climate chamber along with the POFBG sensors for comparison purposes. Both the silica and POFBG 4 (without any applied pre-strain) show a positive and linear Bragg wavelength shift as the temperature increases (Figure 12a). However, when a pre-strain is introduced in POFBG 1-3, the linear thermal expansion is restricted, and the negative thermo-optic coefficient starts to prevail, causing a negative Bragg wavelength shift with increasing temperature (Figure 11). As shown in Figure 12a, the negative response to temperature becomes linear when introducing sufficient fibre pre-strain, such as 0.5% in the case of POFBG 1, which restricts the linear thermal expansion in the whole characterised temperature range. By having insufficient fibre pre-strain, the POF’s thermal expansion is restricted up to a specific temperature. When that temperature point is exceeded, the thermal expansion contribution of the fibre starts to counteract the thermo-optic effect, leading to significantly reduced temperature sensitivity at higher temperatures. As depicted in Figure 12a, POFBG 3 sensor exhibits a negative sensitivity shift up to 40 °C and switches to a relative insensitivity above that temperature. We assume that at a certain temperature point, due to the right amount of fibre tension, the thermal expansion balances the thermo-optic effect, leading to temperature sensing insensitivity. However, well above that temperature point, the thermal expansion prevails over the thermo-optic effect, leading to a non-linear increase of sensitivity to temperature changes.

The calculated sensitivities of silica FBG and POFBG sensors with linear responses are listed in Table 2 for increasing and decreasing temperatures. Comparing Figure 12a with Figure 12b, we note that the sensitivities of POFBG sensors exhibit slight differences during the temperature cycle. A possible reason could be that the thermal expansion of polymers depends not only on the temperature but also on the rate of its change [40]. The difference in heating and cooling rates generates residual strain in the polymer, affecting the Bragg wavelength’s spectral position [40].

## 6. Humidity Characterisation

This section describes the experiment performed to assess the humidity sensitivity of sensors under different pre-strain values. The POFBG sensors were placed in the climate chamber (Memmert HCP 108) to characterise their humidity response over time. The temperature was kept constant at 40 ± 1 °C, and the relative humidity (RH) was changed in steps. The duration of each step was more than 4 h to ensure stabilised climate conditions, maximum water absorption in POFs [12], and minimal readout errors of spectral position of sensors. Figure 13 depicts the Bragg wavelength shift of each sensor over time when the relative humidity changes in steps from 30% to 60% and back to 30% with an increment of 10%. The sensors’ response is relatively linear to increasing (Figure 14a) and decreasing (Figure 14b) relative humidity. Results show that the POFBG 4 (without pre-strain) exhibits the highest humidity sensitivity. The sensitivity can be reduced proportionally by applying fibre pre-strain due to the restriction of swelling [29]. For example, POFBG 1 (with 0.5% pre-strain) becomes insensitive to humidity changes due to the restriction of the water from diffusing into POF. The silica-based sensor responds to some degree to humidity changes because of the acrylate coating surrounding the FBG’s location, which absorbs moisture and physically affects the POFBG structure. Table 3 lists the humidity sensitivities of all sensors calculated with linear fitting.

## 7. Conclusions

In this work, the temperature and relative humidity sensitivities of CYTOP POFBG sensors were characterised under different applied fibre pre-strained values. We report for the first time that the CYTOP POFBG sensors can exhibit positive or negative temperature sensitivities depending on the applied fibre strain. When the POFBG is not under axial tension (strained), the fibre can freely expand due to temperature and water absorption, contributing to the overall Bragg wavelength shift. By introducing an amount of fibre pre-strain, the temperature and humidity sensitivity of POFBG sensors can be tuned to the desired level since the thermal expansion and swelling coefficients are affected by the applied tension. Due to the thermal and thermo-optic properties of CYTOP material, the temperature sensitivity of sensors can be changed from positive to negative, also providing the possibility of obtaining insensitivity to a specific temperature range. The humidity sensitivity of POFBG sensors can also be tuned from positive to zero by introducing a sufficient fibre pre-strain. Without any applied fibre pre-strain, the sensors show positive sensitivity to both temperature and humidity changes. With a specific amount of pre-strain, the sensors are temperature-insensitive but have some degree of humidity sensitivity. By applying a sufficient pre-strain, the sensors exhibit enhanced negative sensitivity to temperature changes but insensitivity to humidity changes. An array of 4 multiplexed POFBG sensors with different temperature and humidity sensitivities is presented in this work for demonstration purposes. The results presented in this work can be considered when designing various POFBG-based sensing systems to tune the sensors’ sensitivities to the desired level for compensation and calibration purposes. For example, using the methodology described in this work, a sensing system of only three sensors can be designed to measure temperature, humidity, and strain/stress; the first sensor can be pre-strained and encapsulated for temperature monitoring only; the second sensor can be used to detect only humidity (some amount of pre-strain) or can be strain-free for simultaneous temperature and humidity measurements; the third sensor can be used to detect all three parameters (temperature, humidity and strain/stress). The temperature measurements from the first sensor can be used to calculate humidity changes detected by the second sensor. Then, the temperature and humidity measurements can be used as compensation to calculate the strain/stress detected by the third sensor.

## Figures and Tables

**Figure 1 sensors-22-07233-f001:**
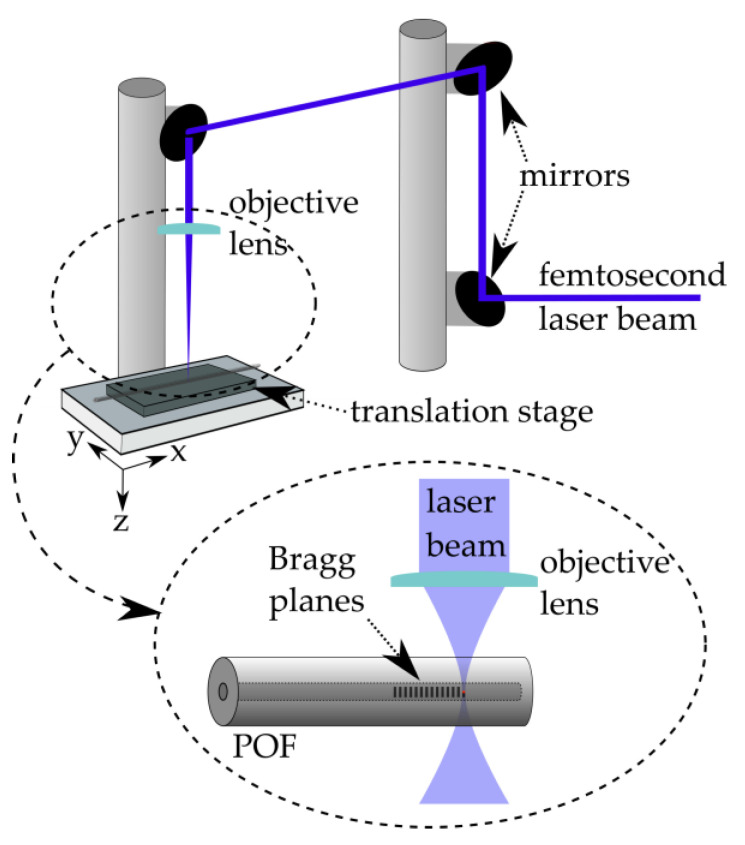
Setup for the fabrication of POFBG sensors.

**Figure 2 sensors-22-07233-f002:**
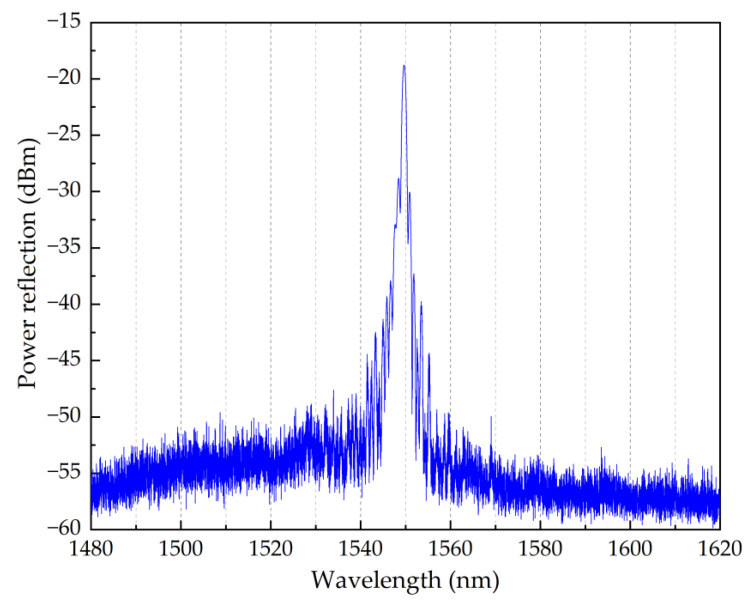
Reflection spectrum of a single POFBG sensor.

**Figure 3 sensors-22-07233-f003:**
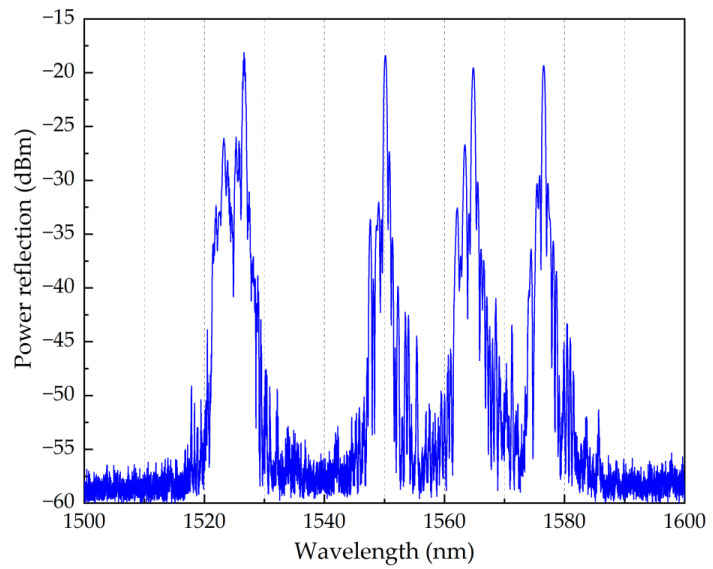
Reflection spectrum of array of 4 POFBG sensors.

**Figure 4 sensors-22-07233-f004:**
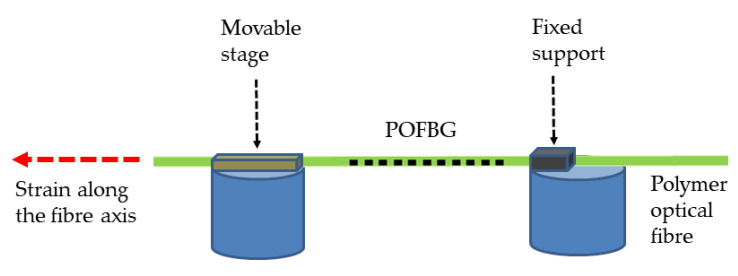
Setup for the strain characterisation of POFBG sensors.

**Figure 5 sensors-22-07233-f005:**
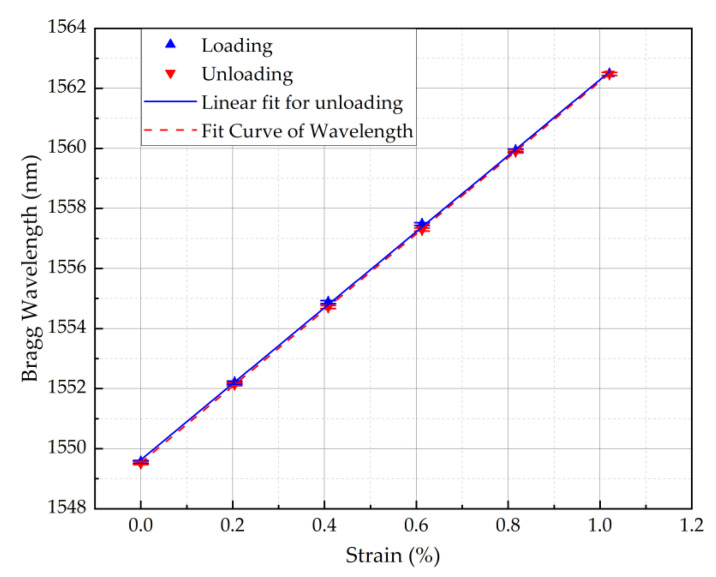
Strain response of POFBG sensor.

**Figure 6 sensors-22-07233-f006:**
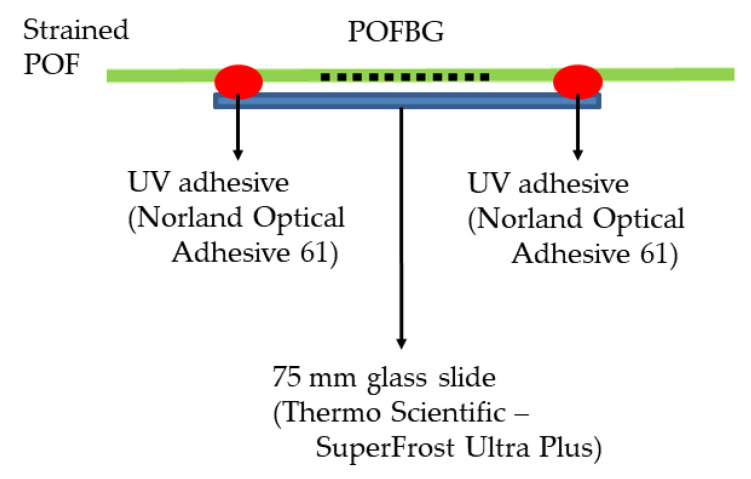
Strained POFBG sensor adhered to the glass slide.

**Figure 7 sensors-22-07233-f007:**
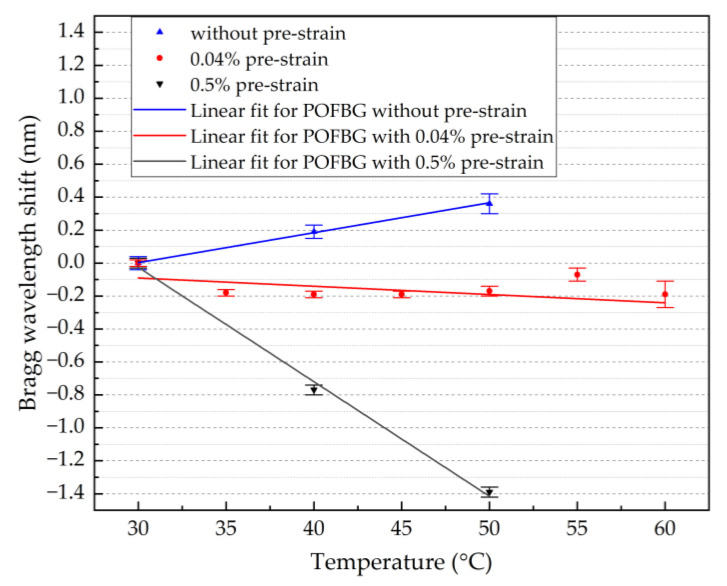
Temperature response of single POFBG under different pre-strain values.

**Figure 8 sensors-22-07233-f008:**
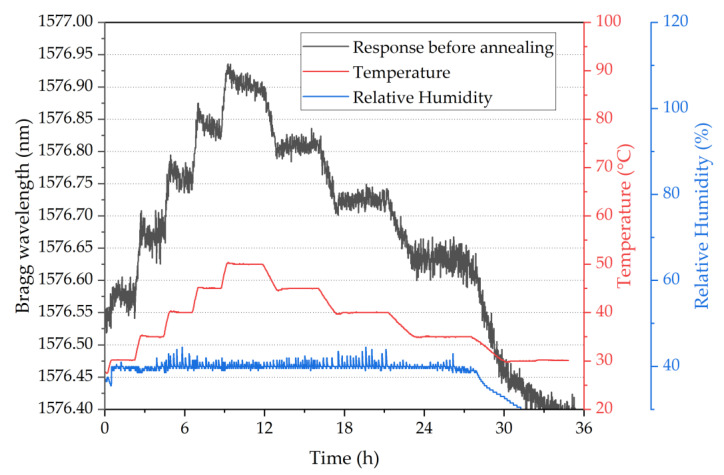
Temperature response of POFBG 4 during the first temperature characterisation.

**Figure 9 sensors-22-07233-f009:**
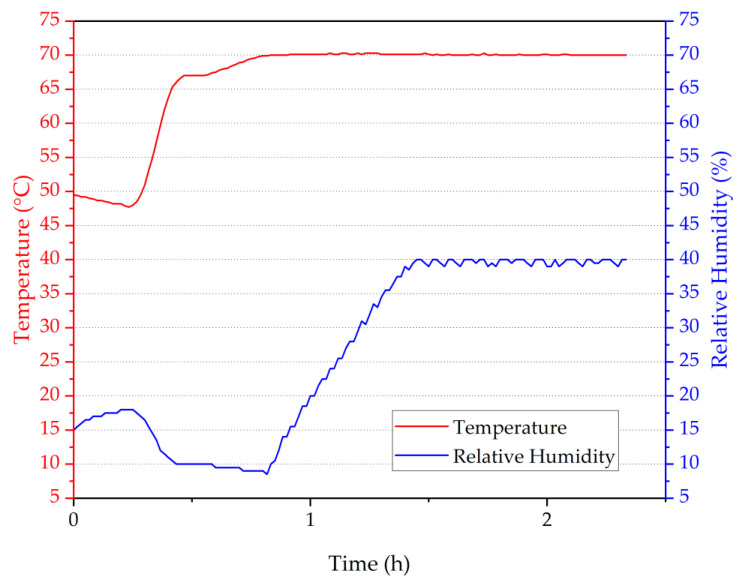
Temperature and humidity profile in the climatic chamber during thermal annealing.

**Figure 10 sensors-22-07233-f010:**
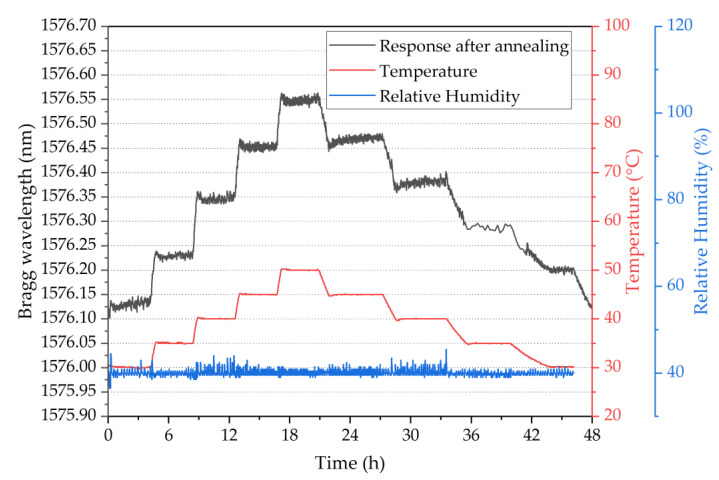
Temperature response of POFBG 4 after thermal annealing treatment.

**Figure 11 sensors-22-07233-f011:**
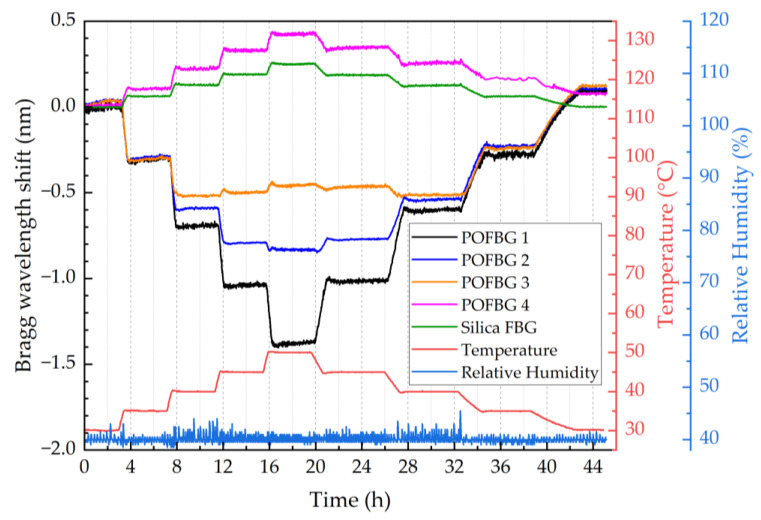
Temperature response of sensors over time.

**Figure 12 sensors-22-07233-f012:**
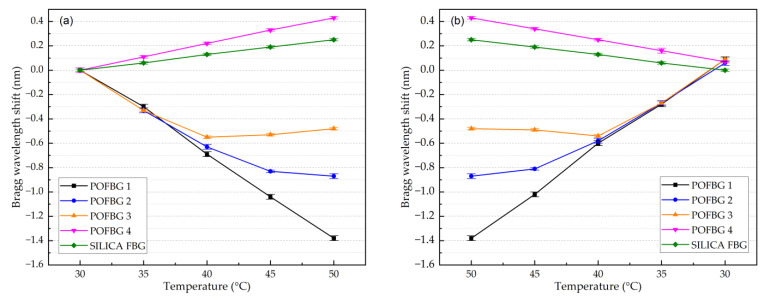
Temperature response of the 4 POFBG array and silica grating sensors under different pre-strain values with (**a**) increasing temperature, and (**b**) decreasing temperature.

**Figure 13 sensors-22-07233-f013:**
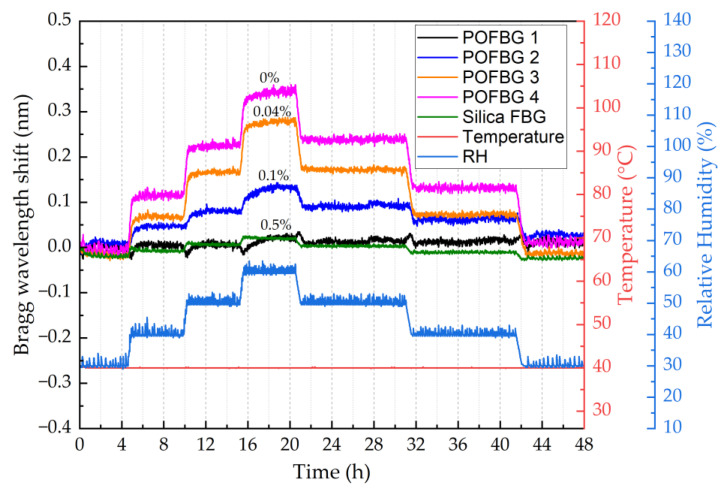
Relative humidity response of sensors over time.

**Figure 14 sensors-22-07233-f014:**
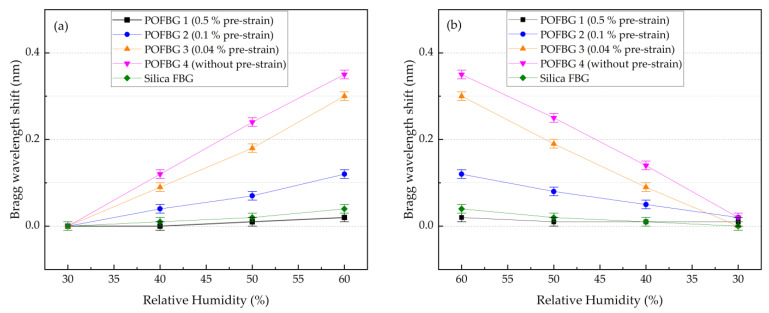
Relative humidity response of the 4 POFBG array and silica grating sensors under different pre-strain values with (**a**) increasing humidity, and (**b**) decreasing humidity.

**Table 1 sensors-22-07233-t001:** Array of POFBG sensors.

Sensor	Bragg Wavelength (nm)	Pre-Strain (%)
POFBG 1	1526.62	0.50
POFBG 2	1550.14	0.10
POFBG 3	1564.79	0.04
POFBG 4	1576.53	0.00

**Table 2 sensors-22-07233-t002:** Temperature sensitivities of sensors.

Sensor	Increasing Temperature	Decreasing Temperature
POFBG 1 (0.5% pre-strain)	−70.0 ± 1.4 pm/°C	−73.6 ± 1.6 pm/°C
POFBG 4 (0% pre-strain)	21.5 ± 0.3 pm/°C	18.0 ± 0.1 pm/°C
Silica FBG (0% pre-strain)	12.6 ± 0.2 pm/°C	12.6 ± 0.2 pm/°C

**Table 3 sensors-22-07233-t003:** Humidity sensitivities of sensors.

Sensor	Increasing Humidity	Decreasing Humidity
POFBG 1 (0.5% pre-strain)	0.7 ± 0.2 pm/%RH	0.3 ± 0.2 pm/%RH
POFBG 2 (0.1% pre-strain)	3.9 ± 0.3 pm/%RH	3.3 ± 0.2 pm/%RH
POFBG 3 (0.04% pre-strain)	9.0 ± 0.5 pm/%RH	10.0 ± 0.3 pm/%RH
POFBG 4 (0% pre-strain)	11.7 ± 0.2 pm/%RH	11.0 ± 0.3 pm/%RH
Silica FBG (0% pre-strain)	1.3 ± 0.2 pm/%RH	1.3 ± 0.2 pm/%RH

## Data Availability

Not applicable.

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
