# Peer review of "Temperature and Humidity Sensitivity of Polymer Optical Fibre Sensors Tuned by Pre-Strain"

_sensors, 2022, doi:10.3390/s22197233_

Round 1

Reviewer 1 Report

In the paper “Temperature and humidity sensitivity of polymer optical fiber sensors tuned by pre-strain”, the authors presented the fabrication and characterization of the Polymer optical fiber Bragg grating sensors. The sensors fabrication process is based on femtosecond laser system in CYTOP optical fibers. The manuscript reports the relative humidity sensitivities of sensors obtained under different pre-strain values.

The manuscript is interesting and the results show the potential for application in multiparameter fiber-optic sensors.

In general, the manuscript is clear, well written and it is suitable for publication in Sensors Journal in the present form.

Author Response

We would like to thank you for reviewing the manuscript and providing positive feedback.

Reviewer 2 Report

Manuscript No:  ms sensors-1923449

Title:  Temperature and humidity sensitivity of polymer optical fibre sensors tuned by pre-strain

Authors:  Andreas Pospori, Andreas Ioannou and Kyriacos Kalli

A. Overview

1. In this manuscript the authors report experimental work on the temperature and humidity response of CYTOP-based POFBG sensors under different fibre pre-strain values.

2. The contents are expressed clearly.

3. The manuscript is well organized,

4. It is written in reasonable English.

5. The authors have acknowledged recent related research.

6. As long as my knowledge, the work presented is original.

7. Several typos along de manuscript, authors must have a second read.

B. Detailed analysis.

Abstract

-Please organize the ideas in each paragraph.

-Be clear, objective.

-State briefly what you did, how did you do it, the quantitative results you and

-State clearly the novelty of your work.

1.    Introduction: provides an interesting approach to the subject and there are up to date references.

2.    Error bares are missing in figures 5, 7, 12, 

C. Overall assessment

The work presented here is very interesting and has potential for further development in the field. In my opinion the manuscript can be published in Sensors, after minor corrections.

D. Review Criteria

1. Scope of Journal

Rating: Medium

2. Novelty and Impact

Rating: Medium

3. Technical Content

Rating: Medium

4. Presentation Quality

Rating: Medium

Author Response

We would like to thank you for reviewing the manuscript and providing your suggestions for improvement.

  1. We have added topic sentences at the start of each section for more clarity. Several other improvements were made according to your recommendations. We have also added in the “Conclusion” the following sentence indicating the novelty of this work: “We report for the first time that the CYTOP POFBG sensors can exhibit positive or negative temperature sensitivity depending on the applied fibre strain.
  2. Two typing errors in measurement units were identified and corrected. We note that the manuscript was written in British English and not in American English. For example, we write “characterisation” instead of “characterization”, “fibre” instead of “fiber”, “normalised” instead of “normalized”, and “stabilised” instead of “stabilized”.
  3. The error bars of plots are more visible now.

Reviewer 3 Report

I believe that this is a rather interesting work that it will be useful to readers and the results of which can be implemented in various fields of science and technology. However, I have a few small remarks that in no way diminish the value of the work.

1. In the introductory parts of the article, the authors mention the issue of the simultaneous influence of various physical factors on the sensor. The authors manufactured the sensors with different sensitivities to different physical influences, but did not draw conclusions about whether such sensors (two of them or more) can discriminate the effects of temperature and humidity, measuring them simultaneously. Instead of this the authors write in the Conclusion: 'The results presented in this work can be considered when designing various POFBG-based sensing systems to tune the sensors’ sensitivities to the desired level for compensation and
calibration purposes.' - i think that this sentence could be extended with the approximate prognosis (as a proposal but not a requirement).

2. Several sensors developed by the authors (POFBG 2 and 3) have a non-linear temperature response. It would be great if the authors gave more detailed hypotheses regarding the reason of this phenomenon.

3. Some sections of this article end with figures or tables. I recommend arranging figures and tables so that they appear after their first mention in the text.

Author Response

We would like to thank you for reviewing the manuscript and providing your suggestions for improvement.

  1. We have added to “Conclusion” the following example to extend the impact statement: “For example, using the methodology described in this work, a sensing system of only three sensors can be designed to measure temperature, humidity, and strain/stress; the first sensor can be pre-strained and encapsulated for temperature monitoring only; the second sensor can be used to detect only humidity (some amount of pre-strain) or can be strain-free for simultaneous temperature and humidity measurements; and the third sensor can be used to detect all three parameters (temperature, humidity and strain/stress). The temperature measurements from the first sensor can be used to calculate humidity changes detected by the second sensor. Then, the temperature and humidity measurements can be used as compensation to calculate the strain/stress detected by the third sensor.”
  2. For the non-linear behaviour of sensors, a further explanation is included in the manuscript: “We assume that at a certain temperature point, due to the right amount of fibre tension, the thermal expansion balances the thermo optic effect, leading to temperature sensing insensitivity. However, well above that temperature point, the thermal expansion prevails over the thermo-optic effect, leading to a non-linear increase of sensitivity to temperature changes.
  3. Figures have been rearranged as you suggested.

Reviewer 4 Report

In this manuscript sensors-1923449, authors studied selectively tuning of polymer optical fibre Bragg grating (POFBG) sensors’ temperature and humidity sensitivities by applying a certain amount of fibre pre-strain. The pre-strain strategy can solve the cross-sensitivity issue of POFBG sensors, since it can maintain the sensitivity to interested parameter while reducing sensitivity to other parameters, so the sensor can measure temperature or humidity more accurately. An array of 4 multiplexed POFBGs were also demonstrated to measure temperature and humidity along the same fibre. Overall, the manuscript is well written, and the introduction, experiment, result and conclusion are well prepared, so I recommend the manuscript been accepted for publishing in Sensors after minor revision. I have some detailed question.

(1)    Could authors give a detailed comparison of PMMA-POFBG and CYTOP-POFBG, e. g. Bragg wavelength shift under different humidity or temperature.

(2)    The fabrication of POFBG with pre-strain on glass will restrict the deformation of POFBG, which will lose the advantage of flexibility compared with silica FBG, have authors considered using other flexible substrates to fabricate POFBG?

(3)    In Figure 10, why the data density from ~36 to ~42h seems lower than other time?

Author Response

We would like to thank you for reviewing the manuscript and providing your suggestions for improvement.

  1. We have added in the manuscript a direct comparison between PMMA and CYTOP POFBG sensors: “It is reported that the temperature sensitivity of PMMA POFBG sensors can be significantly enhanced by applying a sufficient amount of fibre pre-strain to eliminate the counteracting thermal expansion coefficient [33]. PMMA POFBG sensors at 40% relative humidity (RH) exhibit sensitivity ‑37 pm/°C when unstrained and ‑123 pm/°C when pre-strained [33]. At 85% RH, the temperature sensitivity increases to ‑52 pm/°C for the unstrained sensors and −134 pm/°C for the pre-strained sensors [33]. The humidity sensitivity of PMMA POFBG sensors is reported to be 39 pm/%RH at 25°C, 33 pm/%RH at 35°C, and 16 pm/%RH at 50°C [29]. However, pre-straining the PMMA POFBG sensors reduced the humidity sensitivity to 30 pm/%RH at 25°C, 23 pm/%RH at 35°C, and 6 pm/%RH at 50°C [29]. The temperature and humidity sensitivity of CYTOP POFBG sensors is reported to be 27.5 pm/°C and 10.3 pm/%RH, respectively [34]. The pre-strain effects on temperature and humidity sensitivity of CYTOP POFBG sensors were not investigated until now.”
  2. The glass has very low thermal expansion and for this reason it was chosen to place the fibre in order to not contribute to the overall thermal expansion of materials. The fibre is pre-strained (fixed) on the glass, which can be used only to detect temperature and humidity – not mechanical quantities (strain/stress/force/pressure) that require high elasticity. The pre-strain sensors can be used for compensation purposes or in temperature monitoring applications that require higher sensitivity. An example system is added in the conclusion section for further clarification of the purpose of this work.
  3. The data density is different in Figure 10 between 34h and 41h because the 1-minute peak tracking failed to detect the peak in that timeframe (perhaps the peak intensity changed). During the experiment, besides saving the 1-minute peak position, we were also capturing the whole reflection spectrum every 15 minutes. We used this data to generate the missed window. We have added the following statements in the manuscript:
    1. Using a commercial interrogation system (Micron Optics HYPERION si155), the sensors were monitored by tracking the reflected Bragg wavelength peak every minute using the commercial Micron Optics ENLIGHT software. The whole reflection spectrum was also captured every 15 minutes in case the peak tracking failed.
    2. In Figure 10, the data density between 34 h and 41 h is lower because the interrogator failed to detect the peak position (taken every minute) in that timeframe. The missing points have been reconstructed using the data from the reflection spectra taken every 15 minutes.